# Galvanic Effect and Alternating Current Corrosion of Steel in Acidic Red Soil

**Qi-Wei Wang [1], Jun-Xi Zhang [1],\*** , **Yan Gao [1], Nian-Wei Dai [2],\*, Yun-Xiang Chen [3], De-Yuan Lin [3] and Xiao-Jian Xia [3]**

1   Shanghai Key Laboratory of Material Protection and Advanced Material in Electric Power,
    Shanghai University of Electric Power, Shanghai 200090, China; wangqiwei@mail.shiep.edu.cn (Q.-W.W.);
    gaoyan@mail.shiep.edu.cn (Y.G.)
2   CAS Key Laboratory of Mechanical Behavior and Design of Materials, Department of Precision Machinery
    and Precision Instrumentation, University of Science and Technology of China, Hefei 230026, China
3   Electric Power Research Institute of State Grid Fujian Electric Power Company Limited,
    Fuzhou 350007, China; rogerchen614@163.com (Y.-X.C.); lindeyuan_fj@126.com (D.-Y.L.);
    xia.xiaojian@gmail.com (X.-J.X.)
\*   Correspondence: zhangjunxi@shiep.edu.cn (J.-X.Z.); jackienw@ustc.edu.cn (N.-W.D.)

**Abstract:** Alternating current (AC) corrosion behavior of carbon steel–copper couple in acidic red soil was studied by means of the electrochemical test, mass loss, X-ray diffraction (XRD) and scanning electron microscope (SEM) characterization. Mathematical models were established to expound the impacts of AC and galvanic effect on the corrosion mechanism. The results demonstrate that the corrosion rate of the galvanic couple is positively related to AC intensity. Galvanic effect and AC synergistically aggravate the corrosion of steel. The composition of $\alpha$-FeOOH declines while $\gamma$-FeOOH is increased with AC interference. Based on the statistical model, the galvanic effect has a more significant influence on steel corrosion compared with AC.

**Keywords:** galvanic corrosion; alternating current; electrochemical test; corrosion products; mathematical model

## 1. Introduction

The corrosion of the grounding grid under real working conditions has attracted more and more attention since it is crucially important for the safety of a power substation. Several works have reported the effects of sulphate-reducing bacteria (SRB) [1,2], soil resistivity [3,4], the concentration of dissolved oxygen [5,6] and soluble salt ions on the corrosion of metals [7,8]. In addition, AC that passes through the grounding grid also causes serious corrosion of the joints and defects in materials [9]. It is easy for current to escape the intended path and enter the soil to generate stray currents. Compared with the DC (direct current) stray current, the corrosion damage caused by the AC stray current has always been considered to be much weaker. Many previous studies indicated that the cathode branch in one cycle of AC weakened the influence of an anode current [10–13]. However, features of AC leakage are random and intermittent. If high-current AC enters the pipeline, it is likely to cause severe degradation [14,15]. In recent years, corrosion of AC interference on metal materials has spread to many cases of practical engineering. Electrochemical methods and designed circuit systems are used to simulate the influence of AC on metal corrosion behaviour. Theoretical models for corrosion failure caused by AC have been proposed by many researchers. However, there are still some disputes without a unified theoretical explanation. Jones et al. [10] investigated the influence of AC on the corrosion behaviour of carbon steel in 0.1 M NaCl solution. They found that the corrosion rate increased by 4–6 times when 30 mA/cm$^2$ AC was applied to carbon steel. The impact on cathodic polarization is more significant. Kuang et al. [16] researched AC induced pitting corrosion of ×65 pipeline steel in different pH solutions. The results demonstrated

that AC would damage the passive film and promote corrosion of steel. They considered that 30 mA/cm$^2$ was the critical AC density to initiate pitting in the high pH solution, while 20 mA/cm$^2$ was the threshold AC density to initiate pitting in the neutral pH solution.

In actual service conditions, galvanic corrosion will happen when two or more dissimilar metals are coupled with each other. Galvanic corrosion is a common type of corrosion that exists in grounding grids. It usually leads to fracture and perforation of grounding materials. At present, copper-clad steel and galvanized steel are most widely used in grounding grids [17]. Broken copper-clad steel would suffer the coexistence of AC and galvanic effect. For galvanized steel, the zinc coating on galvanized steel is normally damaged and dissolves in 1–3 years, which causes the exposure of bare steel and contact with copper parts. The main factors that affect galvanic corrosion include the ratio of the area between anode and cathode [18,19], the distance between anode and cathode [20,21] and self-corrosion potential difference [22,23]. The current research methods mainly focus on electrochemical impedance spectroscopy (EIS), potentiodynamic polarization curves, zero resistance potential and galvanic current measurement [24,25]. Ni et al. [26] used the scanning vibrating electrode technique (SVET) to study the polarity reversal of Cu–304 stainless steel galvanic couples with different pHs. They concluded that when the pH of the solution reduced from 6 to 0, the polarity reversal of the galvanic couple occurred. The galvanic current was decreased with the reduction of pH. Ikeuba et al. [27] also indicated that galvanic corrosion was related to pH. The corrosion occurred in a localized manner in pH 2 and 6 solutions and was initiated in a uniform manner in pH 13 solutions. In actual engineering practice, as mentioned above, the coexistence of AC interference and galvanic effect is common in grounding grids and causes more serious damage to grounding materials. However, studies regarding the relationship between the galvanic effect and AC on carbon steel corrosion are very limited.

This work aimed to study the contributions of AC and galvanic effect on carbon steel corrosion in acidic red soil. For better implementation of our study, a self-designed indoor simulated exposure experimental device was established with different gradients of AC intensity. According to transmission and transformation conditions, the AC densities were 0, 10, 30, 50, 100 A/m$^2$. Electrochemical measurements and weight loss tests were carried out. Apart from that, SEM and XRD were also used to analyse and characterize the corrosion products. A mathematical model was created and a statistical method was used to understand the contributions of AC interference and galvanic effect in the corrosion process.

## 2. Materials and Methods

### 2.1. Materials and Simulated Solution Preparation

Copper (99.99 wt.%) and carbon steel were used as experimental materials. Samples for weight loss tests were cut into cubes of 20 mm × 10 mm × 5 mm. The chemical compositions of steel are listed in Table 1. All surfaces were polished with 400 grit emery paper, cleaned with deionized water, ultrasonically washed in alcohol and dried in air for 24 h. Thereafter, the specimens were weighed separately with an electronic balance. To ensure the reliability of weight loss, copper wires were welded on a screw on the top of the sample to connect the galvanic couples. The non-working area was sealed with silica gel. The sample is illustrated in Figure 1.

As for electrochemical measurements, the galvanic couple electrode was cut from steel and copper into a 0.5 cm radius semi-circle. The radius of steel and copper electrodes was also 0.5 cm. As Figure 2 shows, all electrodes were sealed with epoxy resin and welded with copper wires on the bottom. The working area of the electrodes was 0.785 cm$^2$. Prepared electrodes were ground with 400–2000 grit emery paper, then polished with alumina paste and washed with deionized water and alcohol.

Acidic red soil that was chosen as the corrosive medium for the weight loss experiment was collected from Fujian Province in the southeast of China. The chemical composition of the soil sample is listed in Table 2.

**Table 1.** Chemical compositions of carbon steel (wt.%).

| C | Mn | Si | S | Fe |
|---|---|---|---|---|
| 0.21 | 0.46 | 0.24 | 0.03 | Bal. |

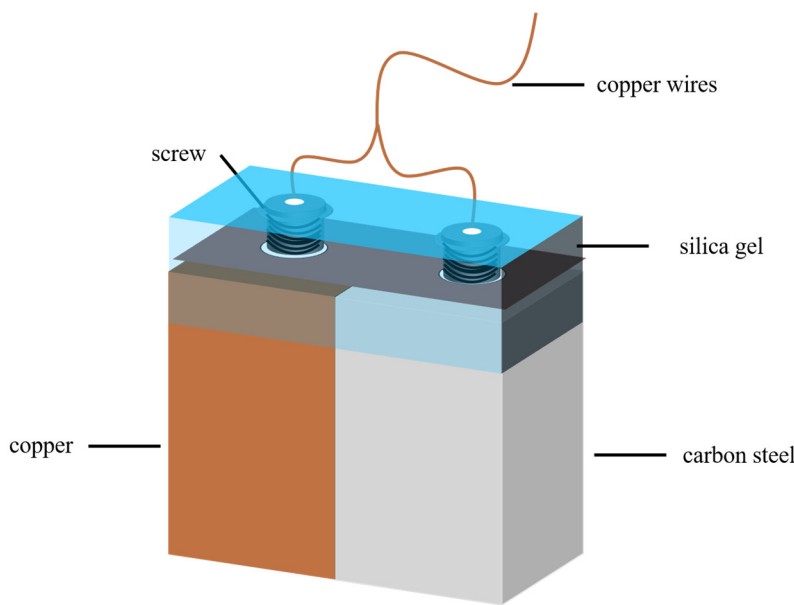

**Figure 1.** Schematic illustration of steel–copper coupled electrode used for weight loss test.

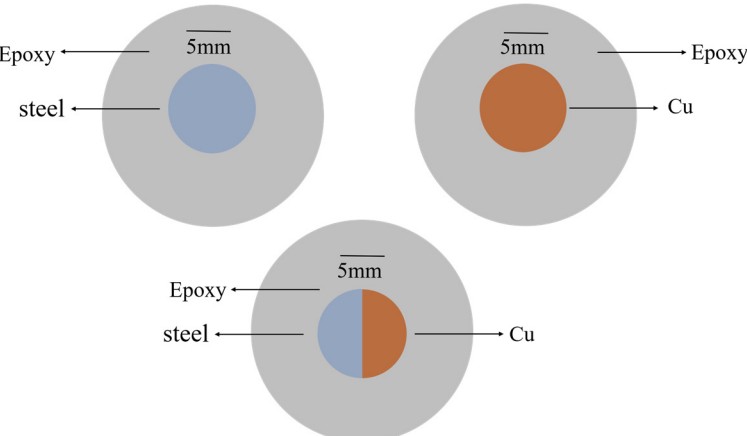

**Figure 2.** Schematic illustration of single steel, copper and coupled steel–copper electrodes used for electrochemical measurements.

**Table 2.** Composition of the experimental soil (wt.%).

| $Cl^-$ | $SO_4^-$ | $HCO_3^-$ | $Ca^{2+}$ | $Na^+$ |
|---|---|---|---|---|
| 0.0084 | 0.0054 | 0.001 | 0.0023 | 0.0032 |

The soils were dried in an oven at 120 °C for 12 h after they were naturally air-dried. Then they were passed through standard 18 mesh sieves. After that, deionized water was added in proportion to obtain soil with a moisture content of 25% (wt.%). The acidic soil simulated solution was prepared for electrochemical measurements according to the properties of the soil. The composition of the acidic soil simulated solution is presented in Table 3. The pH of the simulated solution was adjusted to 4.

**Table 3.** Composition of the simulated acidic soil solution (g·L$^{-1}$).

| NaCl | CaCl$_2$ | Na$_2$SO$_4$ | NaHCO$_3$ |
| --- | --- | --- | --- |
| 0.038 | 0.023 | 0.054 | 0.01 |

### 2.2. Electrochemical Measurements

Electrochemical tests were performed with a conventional three-electrode cell including a platinum counter electrode and a saturated calomel electrode (SCE) as reference. The AC intensity was adjusted by a signal generator (AFG-2225, GW INSTEK, Taipei, China) to 0, 10, 30, 50, 100 A/m$^2$. An inductor (5 H) was connected between the working electrode and electrochemical workstation to avoid AC interference in the electrochemical test system. A capacitor (50 V, 470 μF) was used to isolate the DC signal. A schematic diagram of the equipment is shown in Figure 3.

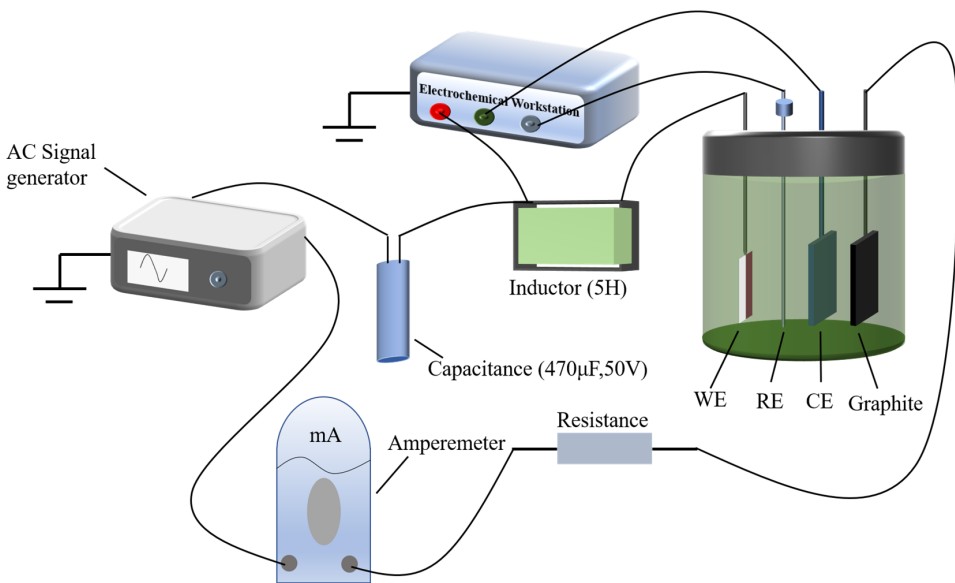

**Figure 3.** Schematic diagram of the electrochemical tests for coupled steel–copper electrodes with interference of different AC intensities in simulated acidic soil solution.

The electrochemical tests, including open circuit potential (OCP) and potentiodynamic polarization tests, were carried out by a DH-7006 electrochemical workstation. Before the polarization tests, the galvanic electrode was put in the simulated solution with AC interference until OCP reached stability. Throughout the measurement process, AC has been applied between the working electrode and the graphite. The scanning range of polarization curves was −0.1 V to 0.1 V versus OCP at the scanning rate of 0.1667 mV/s.

In order to understand the galvanic effect, the electrochemical tests for steel and copper electrodes were also carried out the same as for the galvanic couple electrode. The scanning range of polarization curves for the copper electrode was −1.2 V to 0.1 V versus OCP at the scanning rate of 0.1667 mV/s. The cathodic scanning range of the copper electrode is extended to obtain the data for the galvanic process. All test temperatures were set at 25 ± 2 °C. All experiments were performed at least three times in parallel to ensure the reliability and reproducibility of data. The analysis of polarization curves was performed with cview software by the least squares method.

### 2.3. Weight Loss Tests

A self-designed indoor simulated exposure experimental device was established, as Figure 4 shows. Galvanic couples were buried vertically in the red soil and connected with graphite to form a circuit loop. A rheostat (10–1000 Ω) and capacitor (50 V, 470 μF) were used to form a stable RC loop. A signal generator (AFG-2225, GW INSTEK, Taipei,

China) was used as an AC source. The voltage was adjusted to control AC intensities of 0, 10, 30, 50 and 100 A/m$^2$. At the same time, a group of steel samples without AC interference were used for comparison. A heating pad with a temperature control function was put on the bottom to control the temperature at $25 \pm 2$ °C. AC has been applied to the galvanic couples throughout the experiments. After experiments, silica gel, wires and screws were removed. The surface soil and corrosion products were brushed carefully according to ISO 8407: 202131 [28]. A pickling solution (steel: 500 mL hydrochloric acid, 3.5 g hexamethylenetetramine and 500 mL deionized water; copper: 500 mL hydrochloric acid with 500 mL deionized water) was prepared to clean the remaining corrosion products for 10 min. In order to eliminate the error caused by over-corrosion of the substrate during the rust removal process, a group of blank samples were washed in the same pickling solution. After washing with deionized water and alcohol, all samples were dried for 24 h before being weighed again to calculate the corrosion rate. The formula for the corrosion rate (mm·a$^{-1}$) is as follows.

$$v = \frac{8.7 \times [(w_0 - w_1) - (m_1 - m_2)]}{\rho \times A \times t} \tag{1}$$

where $v$ is the corrosion rate, mm·a$^{-1}$; $w_0$ means the original weight of the Cu–Fe galvanic couple, g; $w_1$ stands for the weight of the sample removing corrosion products, g; $m_1$, $m_2$ is the weight of blank sample before and after cleaning, g; $\rho$ represents the density of steel, g·cm$^{-3}$; $A$ means the working area of the sample, m$^2$; $t$ is the experimental time, h.

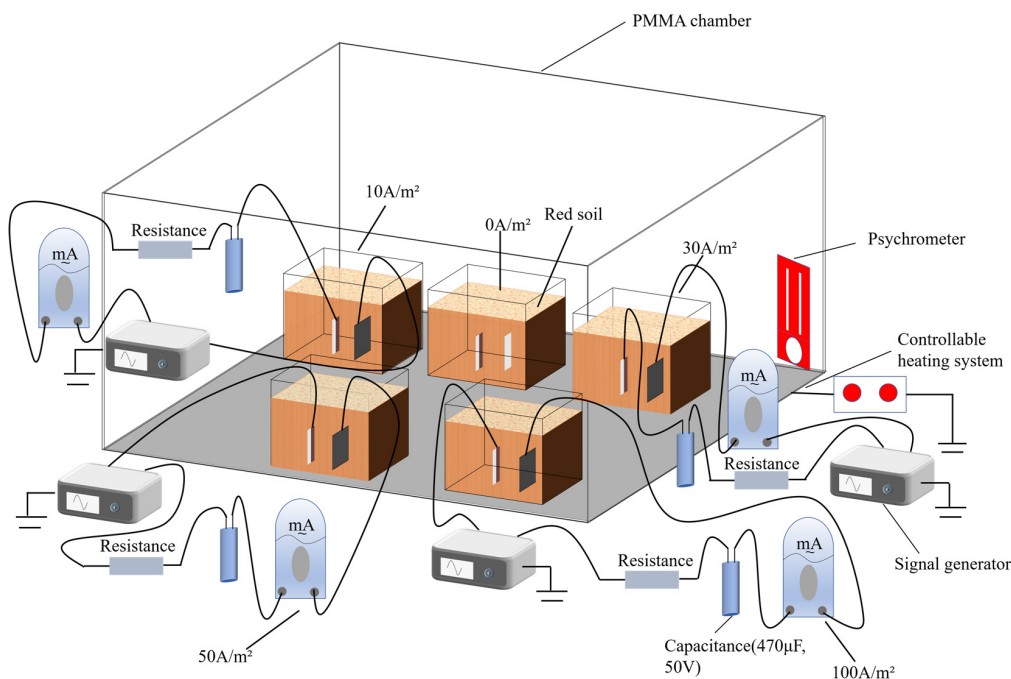

**Figure 4.** Illustration of self-designed chamber to simulate acidic red soil environment with AC interference for weight loss tests.

*2.4. Corrosion Products Characterization*

After weight loss tests, the characterization of the corrosion products of steel in the steel–copper couple with different AC intensities was carried out by XRD (for identification of crystalline phases) and SEM techniques (JSM-7800F, JEOL, Tokyo, Japan) (for microscopic morphology analysis of corrosion products). XRD measurements were carried out with an X-ray diffract meter (D&Advance, Bremen, Germany) with Cu K$_\alpha$ radiation, with $2\theta$ ranging from 10° to 80° at a scan rate of 0.1°/min at room temperature.

*2.5. Statistical Method*

The significance level of AC and galvanic effect to corrosion can be described by the analysis of variance (ANOVA) via the double factor, which considers the influence of two independent variables on numerical dependent variables [29]. The analysis of variance (ANOVA) via the double-factor was conducted by IBM SPSS Statistics software to evaluate the contribution of the galvanic effect and AC to the corrosion process. The AC intensity was factor A, including 0, 10, 30, 50, 100 A/m$^2$ as $A_1$, $A_2$, $A_3$, $A_4$, $A_5$. The galvanic effect was factor B. The corrosion rate of steel is the dependent variable, the AC intensity and galvanic effect are fixed factors. The software was obtained from IBM SPSS Statistics as a subscription trial.

## 3. Results and Discussion

*3.1. Weight Loss and Corrosion Product Analyses*

The weight loss results of the steel–copper couple with different AC intensities in red soil are illustrated in Figure 5, which indicates that the corrosion rate of steel in the galvanic couple is 0.21 mm·a$^{-1}$ without AC interference. There is a positive correlation between the corrosion rate of the galvanic couple and AC. Li et al. [30] researched the corrosion behavior of Q235 steel in Yingtan red soil. They found that the corrosion rate of Q235 steel was nearly 0.1 mm/a. Tang et al. [31] studied the effect of AC current on the corrosion behavior of Q235 steel in acidic soil solution. The result demonstrated that the corrosion rate was nearly 0.2 and 0.25 mm/a when AC density was 50 A/m$^2$ and 100 A/m$^2$. These experimental results demonstrated that the coexistence of AC and galvanic effect accelerated the corrosion of steel. The fitted result demonstrates that the corrosion rate of steel in the galvanic couple calculated by the weight loss data with AC meets the logistic model. The adjusted R$^2$ value of the fitted line is close to 1, which means the precision of the fitted result is high.

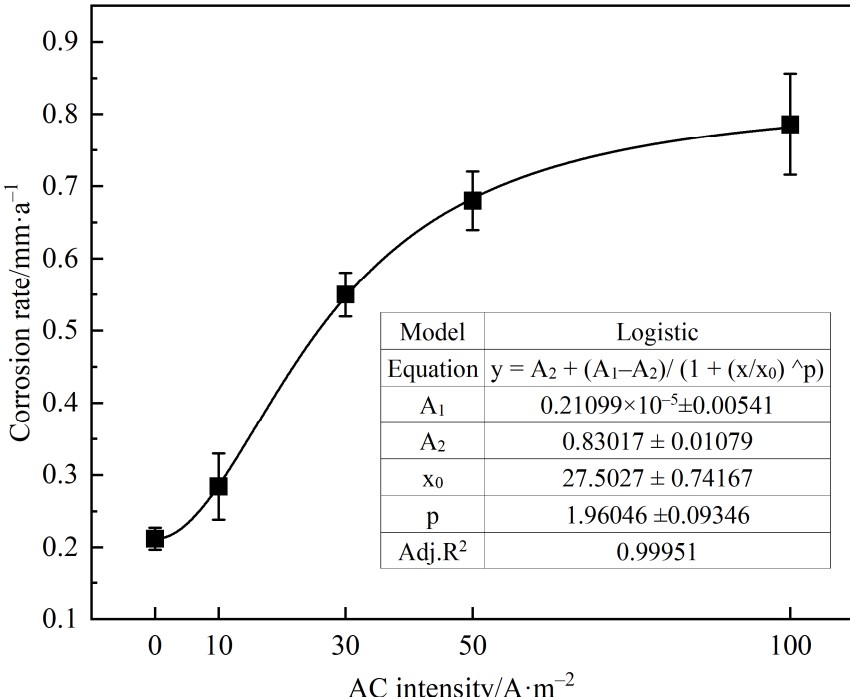

**Figure 5.** Weight loss results of the steel in the galvanic couple interfered with different AC intensities in simulated acidic red soil.

Figure 6 illustrates the corrosion morphology of the steel in the steel–copper couple after the weight loss test. There are fewer surface corrosion products of single steel without AC interference. After coupling with copper and interference with AC, the corrosion

products obviously increase with increasing AC intensity. The characterization of corrosion products develops from localized to uniform corrosion. As Figure 6f shows, the corrosion products completely covered the surface; severe uniform corrosion occurs when the AC intensity is 100 A/m$^2$.

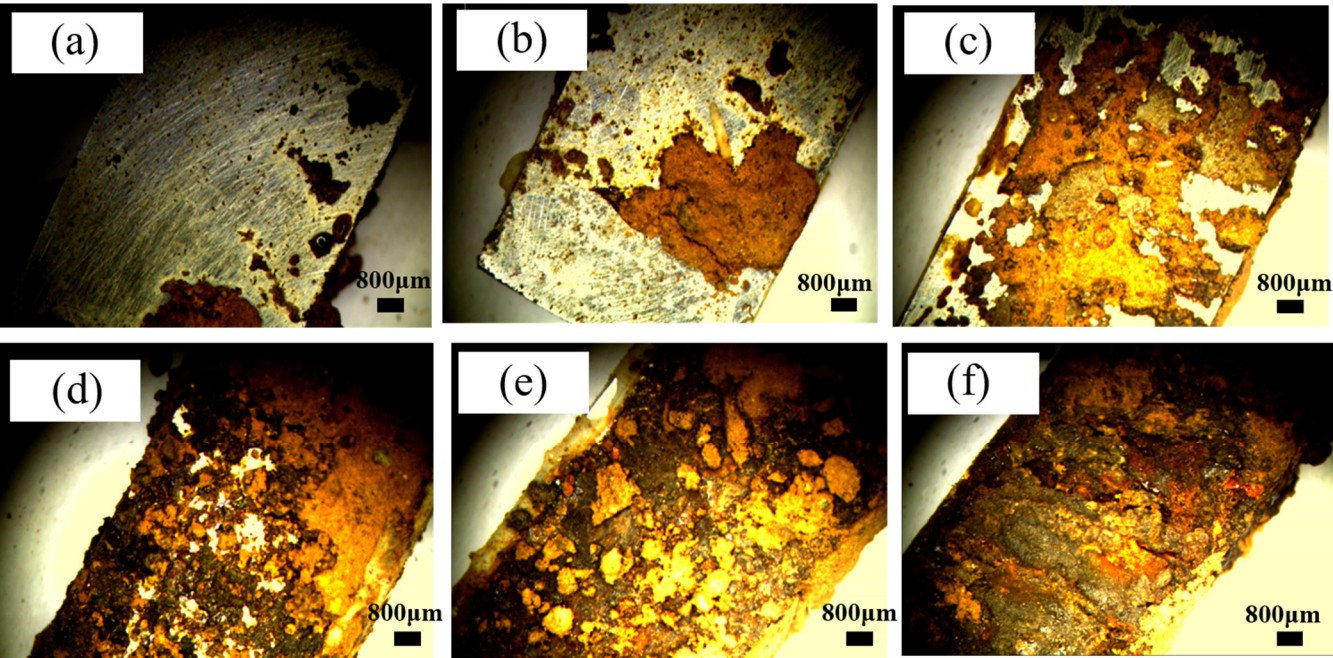

**Figure 6.** Macro corrosion morphology of carbon steel in steel–copper couple after weight loss tests by optical microscope: (**a**) single steel sample without AC interference; (**b–f**) steels in coupled electrodes; AC intensities are 0 A/m$^2$, 10 A/m$^2$, 30 A/m$^2$, 50 A/m$^2$, 100 A/m$^2$, respectively.

In order to study information about the rust in detail, the corrosion products obtained from steel in the steel–copper couple were subjected to X-ray diffraction. The spectra are illustrated in Figure 7. It is clearly seen that the main rust from steel is $\gamma$-FeOOH (JCPDS card no. 18-0639), $\alpha$-FeOOH (JCPDS card no. 26-0792), $\gamma$-Fe$_2$O$_3$ (JCPDS card no. 33-0664) and Fe$_3$O$_4$ (JCPDS card no. 28-0491). SiO$_2$ (JCPDS card no. 47-1144) [32–34] was from the soil. It is generally accepted that the corrosion products of steel mainly include $\alpha$-Fe$_2$O$_3$, Fe$_3$O$_4$, $\alpha$-FeOOH, $\beta$-FeOOH and $\gamma$-FeOOH [35–37]. The value of $\gamma$-FeOOH/$\alpha$-FeOOH from the main intensity peak ratio of 36°/21° for carbon steel in the steel–copper couple is compared with different AC intensities. As Figure 8 shows, "*" represents the single steel sample without AC interference. The value of $\gamma$-FeOOH/$\alpha$-FeOOH is increased when AC applies to the galvanic couple, which indicates that AC inhibits the transformation of $\gamma$-FeOOH to $\alpha$-FeOOH. Many previous works have studied the formation and transformation mechanism of iron oxide [38,39]. The formation and transformation of iron oxides vary in different environments. $\gamma$-FeOOH would transform into $\alpha$-FeOOH and Fe$_2$O$_3$ with a decrease in pH [40,41].

The SEM image of steel in the steel–copper couple after the application of different AC intensities is presented in Figure 9. The shape of the corrosion products is mainly like the cotton ball in Figure 9a,b. Some previous studies [42,43] concluded that $\alpha$-FeOOH is the small and compact cotton-like structure. There are some needle-like structures around the $\alpha$-FeOOH which are $\gamma$-FeOOH. The structure of $\alpha$-FeOOH is stable and compact, while $\gamma$-FeOOH is porous and loose [44]. The corrosion products mainly appeared as porous plate-like structures when AC interfered with the steel–copper couple, which means the composition of $\gamma$-FeOOH is increased with AC interference. A number of plate-like corrosion products are linked together to form a large, dense, porous structure. Such products are multi-hole structures, which led to corrosive ions being able to easily pass

through the rust to reach the substrate. This means that rust cannot protect the metal and the corrosion rate is increased [45].

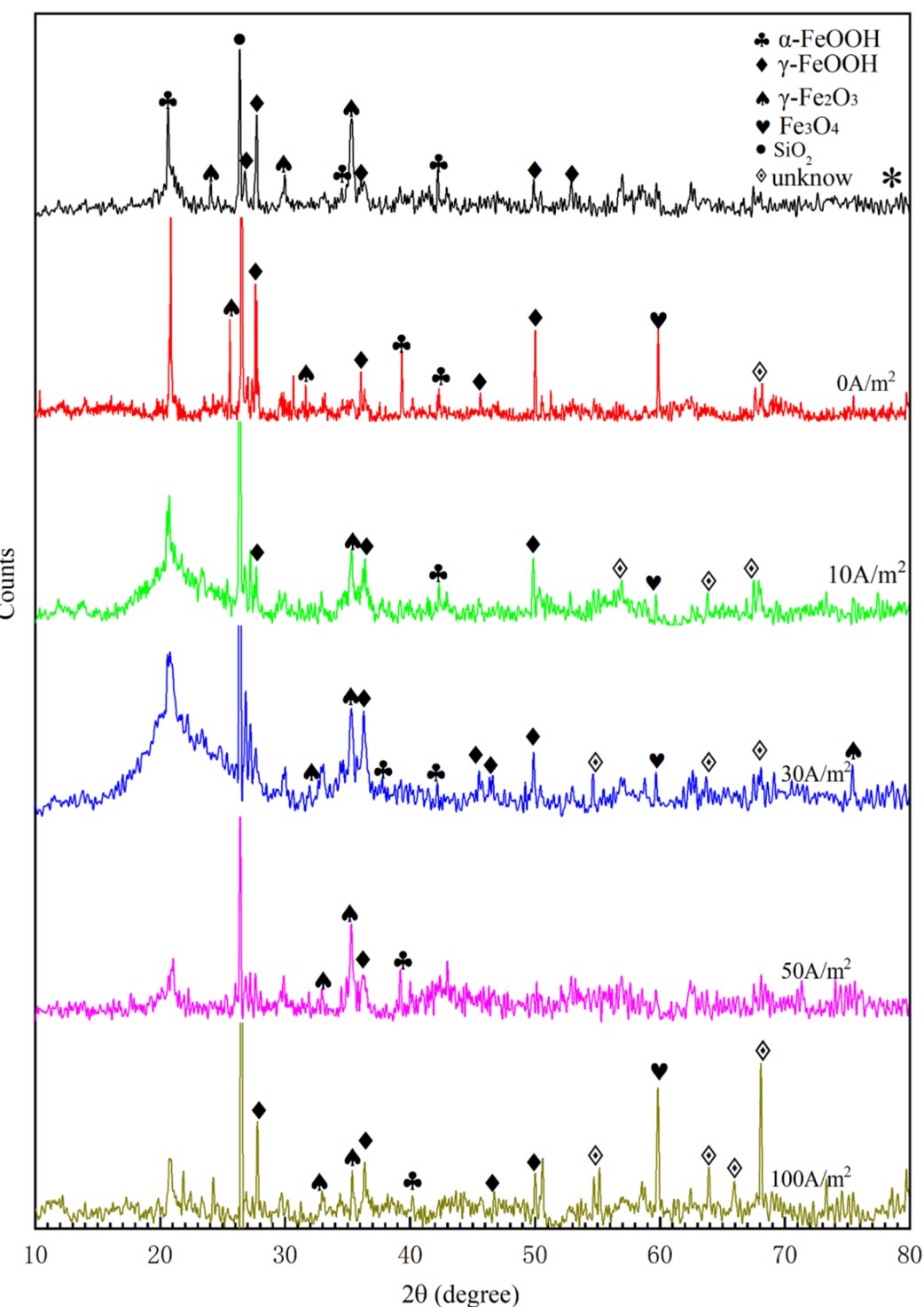

**Figure 7.** XRD patterns of products formed on the steel. * represents single steel sample without AC interference.

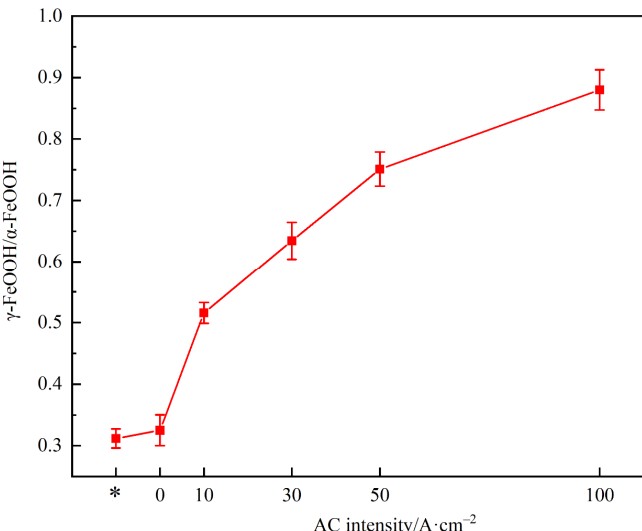

**Figure 8.** Diagram of the value of γ-FeOOH/α-FeOOH for carbon steel in the coupled electrodes subjected to interference with different AC intensities in acidic red soil by XRD patterns. * represents the single steel sample without AC interference.

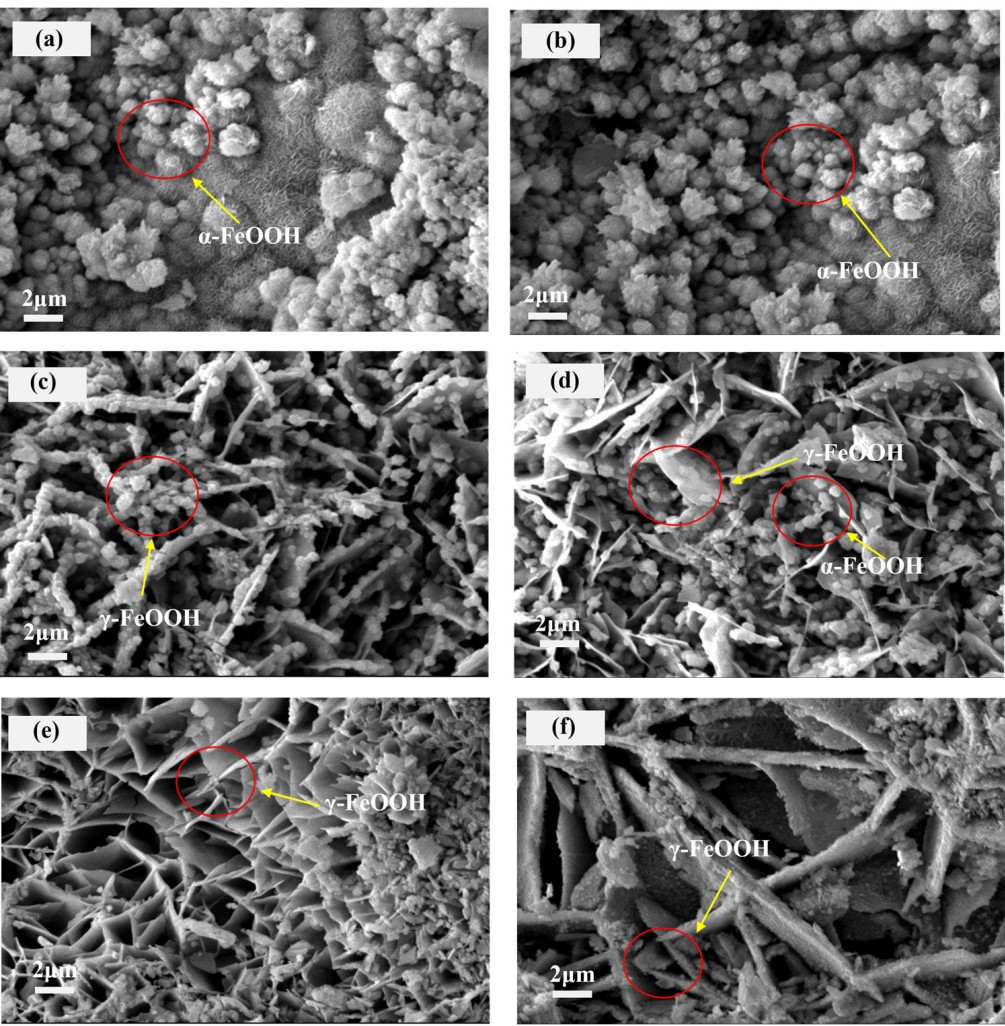

**Figure 9.** SEM images of corrosion products formed on the steel with interference of different AC intensities: (**a**) single steel sample without AC interference; (**b**–**f**) steels in coupled electrodes; AC intensities are 0 A/m², 10 A/m², 30 A/m², 50 A/m², 100 A/m², respectively.

### 3.2. Electrochemical Tests

The fitted corrosion current density ($i_{corr}$) of the steel in the galvanic couple electrode and single steel electrode is shown in Figure 10. The corrosion current density of the specimen increases with the rise of AC interference intensities. The corrosion current density of the steel in the galvanic couple is much higher than single steel with AC disturbance, which means that the galvanic effect accelerates the corrosion process. From the fitted equation of $i_{corr}$ and AC, the $i_{corr}$ of the steel in the galvanic couple and AC obey the logistic model, which is consistent with the result of weight loss. The adjusted $R^2$ value of the fitted lines is 0.991. However, there is a linear relationship between the $i_{corr}$ of the single steel sample and AC. The adjust $R^2$ value of the fitted lines is 0.902, which agrees with the results of Goidanich [15] and Wei [46]. They concluded that the corrosion rate of steel linearly increased with AC in the red soil.

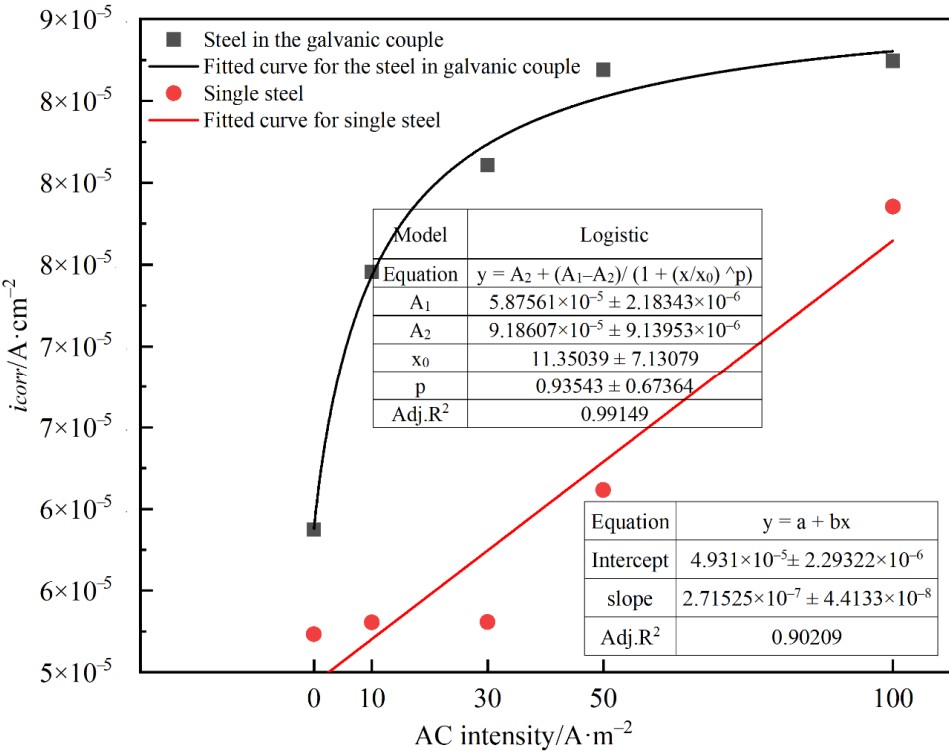

**Figure 10.** The curves of fitted corrosion current density for the steel in the galvanic couple and single steel electrode samples varied with AC interference intensities.

The potentiodynamic polarization result for copper and steel in the acidic soil simulated solution is shown in Figure 11. The analysis of potential and galvanic current is shown in Table 4. The galvanic current is also increased with the rise of AC interference intensities, which means the galvanic corrosion rate is enhanced. The galvanic corrosion rate is the highest at 100 A/m$^2$; the fitted result of the galvanic current with AC is shown in Figure 12, $i_g$ representing the galvanic current. The adjusted $R^2$ value of the fitted line is 0.956. It is also consistent with the logistic model which was brought into correspondence with the fitted model of weight loss and the potentiodynamic polarization results of the steel in the galvanic couple.

**Table 4.** Fitted results of potentiodynamic polarization curve for steel–copper with AC interference.

| $i/\text{A·m}^{-2}$ | 0 | 10 | 30 | 50 | 100 |
|---|---|---|---|---|---|
| $E_g/\text{mV vs SCE}$ | −659.7 | −573.53 | −576.16 | −577.36 | −607.72 |
| $i_g/\text{A·cm}^{-2}$ | $9.0741 \times 10^{-5}$ | $1.3776 \times 10^{-4}$ | $1.7513 \times 10^{-4}$ | $2.0844 \times 10^{-4}$ | $2.2405 \times 10^{-4}$ |

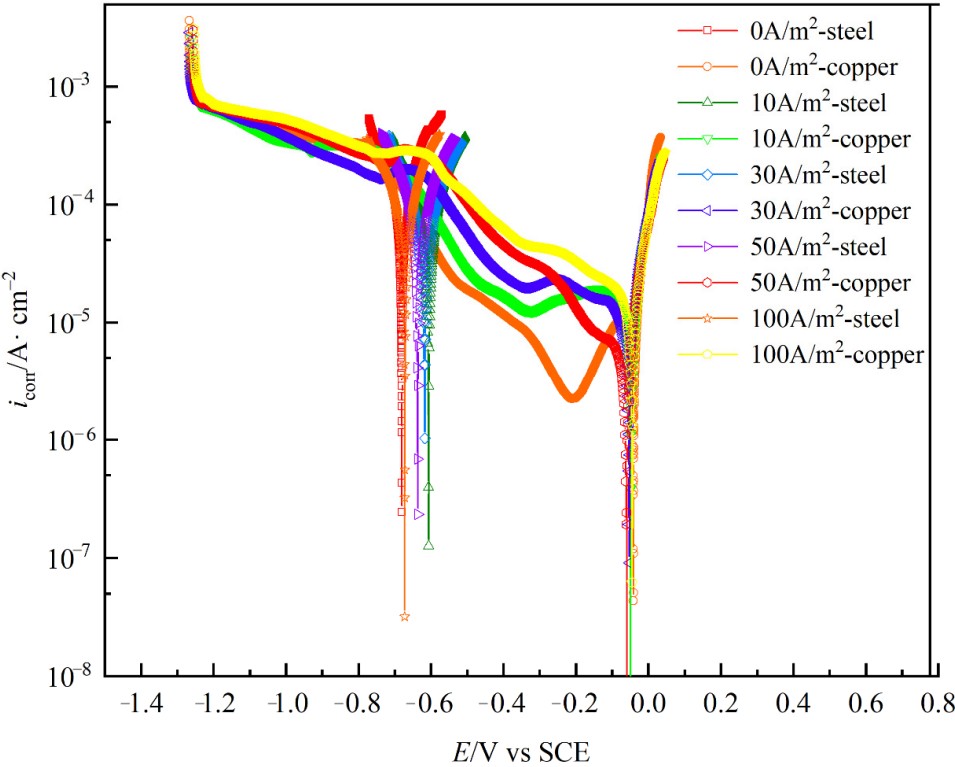

**Figure 11.** Potentiodynamic polarization curves for copper and steel electrodes with interference of AC intensities in simulated acidic soil solution.

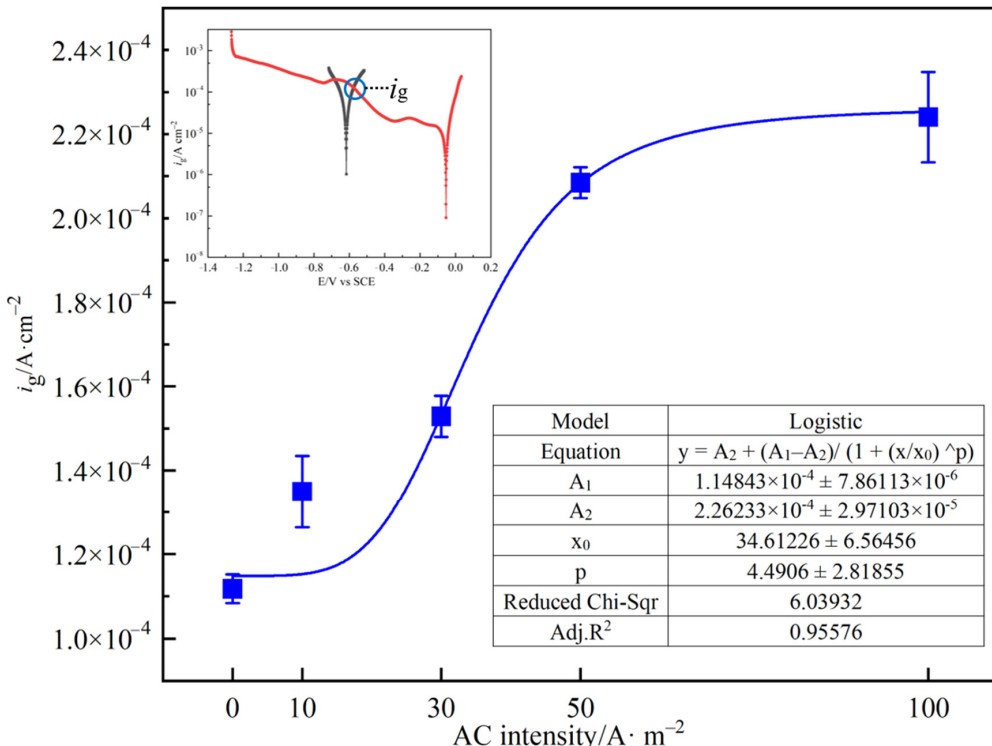

**Figure 12.** The curves of fitted $i_g$ varied with different AC intensities. $i_g$ represents the galvanic current of steel–copper couples and is calculated based on potentiodynamic polarization curves (as shown in inset figure).

### 3.3. AC Corrosion Model for the Galvanic Couple

In order to understand the acceleration effect and contribution of alternating current to corrosion, a mathematical model based on the Tafel equation was built to calculate the effect of AC [47,48]. Assuming that the corrosion process follows the Tafel equation, anode exchange current intensity I0, a remains constant with AC interference when the sample is subjected to AC with the amplitude of $V$ and angular frequency of $\omega$:

$$I_a = I_{corr} exp\left[\frac{Vsin(\omega t)}{\beta_a}\right] \tag{2}$$

$\beta_a$ is the anode Tafel slope.

To $exp\left[\frac{Vsin(\omega t)}{\beta_a}\right]$, for the first kind of modified Bessel function expansion:

$$exp\left[\frac{V}{\beta_a}sin(\omega t)\right] = J_0\left(\frac{V}{\beta_a}\right) + 2\sum_{k=0}^{\infty}(-1)^k J_{2k+1}\left(\frac{V}{\beta_a}\right)sin[(2k+1)\omega t] + 2\sum_{k=1}^{\infty}(-1)^k J_{2k}\left(\frac{V}{\beta_a}\right)cos(2k\omega t) \tag{3}$$

$J_n(x)$ is the first modified Bessel function of order $n$,

$$J_n(x) = \sum_{r=0}^{\infty}\frac{\left(\frac{x}{2}\right)^{n+2r}}{r!(n+r)!} \tag{4}$$

The anodic dissolution current density of the metal under the amplitude V is as follows:

$$I_a = I_{corr}\left\{J_0\left(\frac{V}{\beta_a}\right) + \sum_{k=0}^{\infty}(-1)^k J_{2k+1}\left(\frac{V}{\beta_a}\right)sin[(2k+1)\omega t] + \sum_{k=1}^{\infty}(-1)^k J_{2k}\left(\frac{V}{\beta_a}\right)cos(2k\omega t)\right\} \tag{5}$$

In order to obtain the average of I, t is integrated in a period T (T = 1/f) and over the period T. Since $sin[(2k+1)\omega t]$ and $cos(2k\omega t)$ are periodic functions, the integral value is 0. $I_{corr}J_0\left(\frac{V}{\beta_a}\right)$ is the DC effect caused by AC, which is related to the corrosion rate of metals.

$$(I_a)_{dc} = I_{corr}J_0\left(\frac{V}{\beta_a}\right), \quad J_0\left(\frac{V}{\beta_a}\right) = \sum_{r=0}^{\infty}\frac{\left(\frac{V}{2\beta_a}\right)^{2r}}{(r!)^2} \tag{6}$$

According to Formula (6), the first five terms ($r = 4$) are used to estimate the effect of AC induction. The anodic dissolution rate of galvanic couples is, respectively, 1.085, 1.143, 1.18 and 1.239 times larger than without AC interference of 10, 30, 50 and 100 A/m². As Figure 13 shows, $n$ represents the ratio of the anodic dissolution rate of the galvanic couple electrodes with and without AC interference. It shows a similar configuration to the curves of fitted corrosion current density for the steel–copper couples. The fitted equation also indicates that there is a logistic relationship between $n$ and AC. The adjust $R^2$ value of the fitted line is 0.976, which is also consistent with the fitted model of the above results.

### 3.4. Variance Analysis Based on ANOVA

Some previous studies have introduced ANOVA via the double-factor model to analyse the corrosion issue [49,50]. The results are highly reliable. With regard to our study, as mentioned above, AC intensity and galvanic effect are regarded as the fixed factors; the corrosion rate of steel is the dependent variable. The results are listed in Table 5. It is apparent that the significance value of AC, the galvanic effect and interactions are less than 0.05 with $\alpha = 0.05$, which means that they all have a great influence on corrosion. The effects of interactions between factors were not taken into account in these relationships. However, the F value of the galvanic effect is 217.733 higher than the AC of 62.707. The result demonstrates that the galvanic effect and AC promoted corrosion, but the galvanic effect is more significant than AC in the corrosion process. This is consistent with the above results.

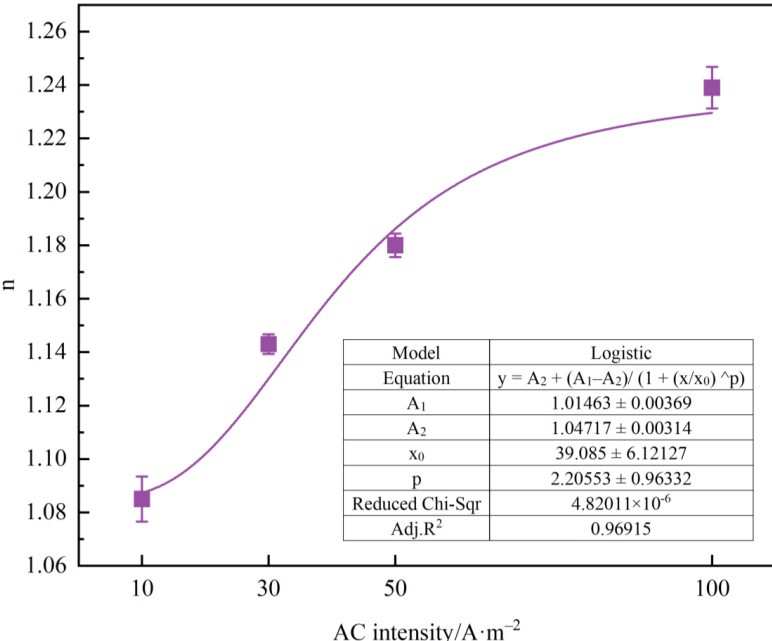

**Figure 13.** The curves for n varied with different AC intensities. n represents the ratio of the anodic dissolution rate of the galvanic couple electrodes with and without AC interference calculated by the AC corrosion model.

**Table 5.** Results based on the analysis of variance.

| Source | Three Types of Sums of Squares | Degree of Freedom | Mean Square | F | Significance |
|---|---|---|---|---|---|
| Revised model | 5947.060$\alpha$ | 9 | 660.784 | 56.751 | 0.000 |
| Intercept | 134,526.792 | 1 | 134,526.792 | 11,553.836 | 0.000 |
| Galvanic effect | 2535.172 | 1 | 2535.172 | 217.733 | 0.000 |
| AC | 2920.531 | 4 | 730.133 | 62.707 | 0.000 |
| Error | 232.869 | 20 | 11.643 | \ | \ |
| Total | 140,706.722 | 30 | \ | \ | \ |
| Revised total | 6179.929 | 29 | \ | \ | \ |

$\alpha$ = 0.05 of this study.

### 3.5. Corrosion Mechanism of Coupled Steel–Copper Electrodes with AC Interference

Through electrochemical measurements, the weight loss test and the characterization of corrosion products, as well as the establishment of mathematical and statistical models, it is obvious that the galvanic effect and AC interference caused a synergistic effect on the corrosion of steel. When AC was not interfering with the galvanic couple, the steel is the anode while the copper is the cathode of the galvanic couple. The reactions of anode and cathode are as follows:

$$\text{Fe} \rightarrow \text{Fe}^{2+} + 2\text{e}^- \tag{7}$$

$$\text{O}_2 + 2\text{H}_2\text{O} + 4\text{e}^- \rightarrow 4\text{OH}^- \tag{8}$$

$\text{Fe}^{2+}$ is hydrolysed:

$$\text{Fe}^{2+} + 2\text{H}_2\text{O} \rightarrow \text{Fe(OH)}_2 + 2\text{H}^+ \tag{9}$$

$\text{Fe(OH)}_2$ is unstable and further reacts with $\text{O}_2$:

$$4\text{Fe(OH)}_2 + \text{O}_2 \rightarrow 4\text{FeOOH} + 2\text{H}_2\text{O} \tag{10}$$

$\gamma$-FeOOH is also unstable and reacts with $\text{Fe}^{2+}$:

$$\text{FeOOH} + \text{Fe}^{2+} + 2\text{e}^- \rightarrow \text{Fe}_3\text{O}_4 + 2\text{H}_2\text{O} \tag{11}$$

As is indicated in Figure 14a, the anodic dissolution of steel is accelerated when coupled with copper. As a current loop of the galvanic corrosion process, more $Fe^{2+}$ escapes from steel, $H^+$ migrates to the cathode area, while the negative ion, such as $SO4^{2-}$ and $OH^-$, migrate to the anode area. When AC is imposed on the galvanic electrode, as Figure 14b shows, the positive branch of AC increases the anode dissolution of electrochemical processes. $H^+$ is pulled away from the electrode/electrolyte interface. The hydrolysis of $Fe^{2+}$ is promoted, which causes more $Fe^{2+}$ to dissolve to the electrolyte. Anode dissolution is further accelerated. During the negative branch of AC, as exhibited in Figure 14c, the reduction of oxygen is facilitated and enhanced. $SO4^{2-}$ and $OH^-$ ions are pulled away from the electrode/electrolyte interface.

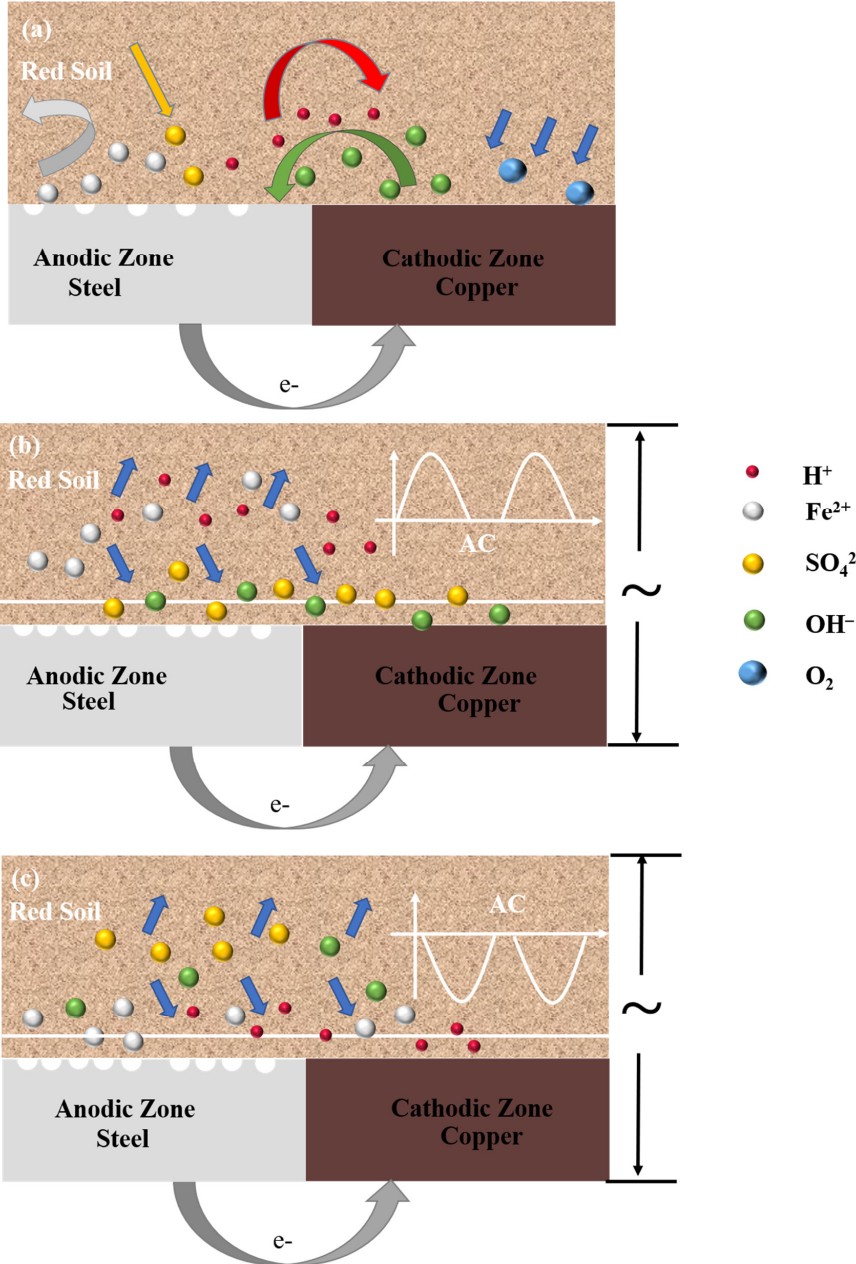

**Figure 14.** Schematic of corrosion processes and mechanism: (**a**) the galvanic couple without AC; (**b**) the galvanic couple with the positive branch of AC; (**c**) the galvanic couple with the negative branch of AC. Arrows represent migration direction of particles.

The formation and transformation of Fe oxide are important corrosion processes for steels. Some previous studies [51,52] have concluded that $\gamma$-FeOOH transforms to $\alpha$-FeOOH by dissolution of $\gamma$-FeOOH and precipitation of $\alpha$-FeOOH with the coexistence of $SO_4^{2-}$ and $Fe^{2+}$. In the positive branch of the AC signal, $Fe^{2+}$ and $H^+$ shift out of the electrode/electrolyte interface while $SO_4^{2-}$ pulls to the electrode/electrolyte interface. In the negative branch of AC, $SO_4^{2-}$ migrates out of the electrode/electrolyte interface while $Fe^{2+}$ immigrates into the interface. Therefore, the coexistence of $Fe^{2+}$ and $SO_4^{2-}$ is broken with AC interference, which may hinder the transformation of $\gamma$-FeOOH to $\alpha$-FeOOH. The structure of $\gamma$-FeOOH is loose and porous and so cannot protect the metal. Corrosion is further promoted. According to some studies, the formation free energy of $\alpha$-FeOOH (496 kJ/mol) is higher than $\gamma$-FeOOH (471 kJ/mol) [53,54]. This is in good agreement with the XRD result. With regard to our study, when AC is applied to the galvanic couple, the growth of $\gamma$-FeOOH is prior to $\alpha$-FeOOH. The existence of AC not only inhibits the transformation of $\gamma$-FeOOH to $\alpha$-FeOOH but also promotes the growth of $\gamma$-FeOOH.

### 4. Conclusions

In this paper, the corrosion behaviour of the Cu–Fe galvanic couple in acidic red soil with AC interference was studied by simulated exposure experiments, electrochemical tests and mathematical and statistical models. The main conclusions are as follows:

(1). Based on the results of electrochemical and mass loss experiments, the corrosion rate of the steel in the steel–copper couple is increased by increasing the AC intensity in a relatively monotonic manner, reaching the maximum value when the applied AC intensity is 100 A/m$^2$.

(2). The existence of AC changes the ion migration of the galvanic couple, which inhibits the transformation of $\gamma$-FeOOH to $\alpha$-FeOOH and promotes the growth of $\gamma$-FeOOH.

(3). The galvanic effect and AC interference cause a synergistic effect on the corrosion of steel in the steel–copper couple. The AC corrosion of steel is further deteriorated when coupled with copper.

(4). Through the ANOVA via the double-factor, it reveals that the galvanic effect is much more significant than the AC aspect in carbon steel corrosion.

**Author Contributions:** Data curation, writing—original draft, Q.-W.W.; conceptualization, methodology, N.-W.D.; supervision, J.-X.Z.; investigation, Y.-X.C. and D.-Y.L.; review, X.-J.X.; visualization, Y.G. All authors have read and agreed to the published version of the manuscript.

**Funding:** This work was supported by the National Natural Science Funds of China (No. 52171074). This work was also supported by the Science and Technology Commission of Shanghai Municipality (No. 19DZ2271100).

**Institutional Review Board Statement:** Not applicable.

**Informed Consent Statement:** Not applicable.

**Data Availability Statement:** The data that support the findings of this study are available from the corresponding author upon reasonable request.

**Conflicts of Interest:** The authors declare no competing financial interest.

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
