# Peer review of "Galvanic Effect and Alternating Current Corrosion of Steel in Acidic Red Soil"

_metals, doi:10.3390/met12020296_

Round 1

Reviewer 1 Report

The paper entitled “ Galvanic Effect and Alternating Current Corrosion of Steel in Acidic Red Soil” focuses on contributions of AC and galvanic effect on the carbon steel corrosion in acidic red soil. The carbon steel-copper couple performance was evaluated by electrochemical test, mass loss, X-ray diffraction (XRD), and scanning electron microscope (SEM) characterization. Mathematical and statistical models are used to understand the acceleration effect and contribution of alternating current to corrosion and corrosion rate in dependence on AC intensity and galvanic effect, respectively.

Тhe introduction refers to the aim of the study, the experimental part is consistently revealed and explained while the results are understandably submitted and sufficiently illustrated. The conclusion summarizes the aforementioned results. In my opinion, the paper should be interesting from a scientific and practical point of view.

I would like to recommend the publication of paper publication after some changes concerning the following issues:  

  1. The magnification in Fig. 6 is unclear. Please, make it visible;
  2. It is not explained how the authors have chosen the levels of the adjacent AC density levels namely 0, 10, 30, 50, 100 A/m2. Would it not be reasonable for these levels to be at regular intervals.
  3. The JCPDS No cards should be added to the identified phase to increase the reproducibility of the results.
  4. In the XRD patterns, there are unidentified phases. Some uncommented amorphous phases are also present on the surface of steels in coupled electrodes treated with AC intensities of 10 A/m2 and 30 A/m2
  5. Fig. 11 should be re-plotted for better visualization of the results. For instance, “Cu” and “Fe” could be presented on separate graphs;
  6. Where applicable, the obtained results can be compared with other similar studies;
  7. Please, pay attention to the indices and powers in the text, figures and tables;
  8. In some places, the English style and grammar have to be improved. Some examples of inadequate style: “w1 stans for the weight of the sample removing corrosion products”; “The main factors that affecting galvanic corrosion include the ratio of….” …and some more….

Reviewer 2 Report

The manuscript, entitled „Galvanic Effect and Alternating Current Corrosion of Steel in Acidic Red Soil” is relevant to the scope of this journal.

It is an interesting study that can provide interesting information to specialists.

The authors have made a good synthesis of the literature that provides an overview of the research evolution in this area.

However, some points need to be addressed prior to publication of this manuscript. My comments/suggestions are given:

  1. In the chapter on materials and methods, the authors must complete subchapter 2.4. More data on the methods used to characterize corrosion products need to be added.
  2. I do not understand the unit of measurement for the corrosion rate of the galvanic couple “0.21 mm a -1”. Please check and explain.
  3. More comments should be added regarding the curve shown in Figure 5. What type of curve is it, etc. Comparison with existing data in the literature is mandatory!
  4. It is necessary to specify the type of images from Figure 6, optical micrographs, SEM…
  5. The scale must be added to each image from Figure 6.
  6. For XRD analysis, the cards used to identify the products presented must be specified.
  7. In the legend of Figure 7 it must be very clear what each spectrum means (conditions or for what sample they are).
  8. Figure 8 refers to the value of γ-FeOOH / α-FeOOH. What size is it, how was it determined and quantified? Please specify to be clearer.
  9. Please specify how you determined the corrosion current density! Analyzing Figure 11, it seems to me that icorr for steel has higher values than those shown in Figure 10.
  10. In the legend of Figure 11, Fe should be replaced with steel, because it has been studied!

Round 2

Reviewer 1 Report

The authors have carefully addressed all the review’s recommendations and the manuscript has been substantially improved. 

Reviewer 2 Report

The authors made all the required corrections, so the manuscript has been greatly improved. Thus, the manuscript may be published in its present form.